# FAIR ATTRIBUTE COMPLETION ON GRAPH WITH MISSING ATTRIBUTES

**Dongliang Guo[1], Zhixuan Chu[2], Sheng Li[3,1]**
[1]University of Georgia, [2]Ant Group, [3]University of Virginia
`dg54883@uga.edu, chuzhixuan.czx@alibaba-inc.com, shengli@virginia.edu`

## ABSTRACT

Tackling unfairness in graph learning models is a challenging task, as the unfairness issues on graphs involve both attributes and topological structures. Existing work on fair graph learning simply assumes that attributes of all nodes are available for model training and then makes fair predictions. In practice, however, the attributes of some nodes might not be accessible due to missing data or privacy concerns, which makes fair graph learning even more challenging. In this paper, we propose FairAC, a fair attribute completion method, to complement missing information and learn fair node embeddings for graphs with missing attributes. FairAC adopts an attention mechanism to deal with the attribute missing problem and meanwhile, it mitigates two types of unfairness, i.e., feature unfairness from attributes and topological unfairness due to attribute completion. FairAC can work on various types of homogeneous graphs and generate fair embeddings for them and thus can be applied to most downstream tasks to improve their fairness performance. To our best knowledge, FairAC is the first method that jointly addresses the graph attribution completion and graph unfairness problems. Experimental results on benchmark datasets show that our method achieves better fairness performance with less sacrifice in accuracy, compared with the state-of-the-art methods of fair graph learning. Code is available at: `https://github.com/donglgcn/FairAC`.

## 1 INTRODUCTION

Graphs, such as social networks, biomedical networks, and traffic networks, are commonly observed in many real-world applications. A lot of graph-based machine learning methods have been proposed in the past decades, and they have shown promising performance in tasks like node similarity measurement, node classification, graph regression, and community detection. In recent years, graph neural networks (GNNs) have been actively studied (Scarselli et al., 2008; Wu et al., 2020; Jiang et al., 2019; 2020; Zhu et al., 2021c;b;a; Hua et al., 2020; Chu et al., 2021), which can model graphs with high-dimensional attributes in the non-Euclidean space and have achieved great success in many areas such as recommender systems (Sheu et al., 2021). However, it has been observed that many graphs are biased, and thus GNNs trained on the biased graphs may be unfair with respect to certain sensitive attributes such as demographic groups. For example, in a social network, if the users with the same gender have more active connections, the GNNs tend to pay more attention to such gender information and lead to gender bias by recommending more friends to a user with the same gender identity while ignoring other attributes like interests. And from the data privacy perspective, it is possible to infer one's sensitive information from the results given by GNNs (Sun et al., 2018). In a time when GNNs are widely deployed in the real world, this severe unfairness is unacceptable. Thus, fairness in graph learning emerges and becomes notable very recently.

Existing work on fair graph learning mainly focuses on the pre-processing, in-processing, and post-processing steps in the graph learning pipeline in order to mitigate the unfairness issues. The pre-processing approaches modify the original data to conceal sensitive attributes. Fairwalk (Rahman et al., 2019) is a representative pre-processing method, which enforces each group of neighboring nodes an equal chance to be chosen in the sampling process. In many in-processing methods, the most popular way is to add a sensitive discriminator as a constraint, in order to filter out sensitive information from original data. For example, FairGNN (Dai & Wang, 2021) adopts a sensitive

classifier to filter node embeddings. CFC (Bose & Hamilton, 2019) directly adds a filter layer to deal with unfairness issues. The post-processing methods directly force the final prediction to satisfy fairness constraints, such as (Hardt et al., 2016).

When the graphs have complete node attributes, existing fair graph learning methods could obtain promising performance on both fairness and accuracy. However, in practice, graphs may contain nodes whose attributes are entirely missing due to various reasons (e.g., newly added nodes, and data privacy concerns). Taking social networks as an example, a newly registered user may have incomplete profiles. Given such incomplete graphs, existing fair graph learning methods would fail, as they assume all the nodes have attributes for model training. Although FairGNN (Dai & Wang, 2021) also involves the missing attribute problem, it only assumes that a part of the sensitive attributes are missing. To the best of our knowledge, addressing the unfairness issue on graphs with some nodes whose attributes are entirely missing has not been investigated before. Another relevant topic is graph attribute completion (Jin et al., 2021; Chen et al., 2020). It mainly focuses on completing a precise graph but ignores the unfairness issues. In this work, we aim to jointly complete a graph with missing attributes and mitigate unfairness at both feature and topology levels.

In this paper, we study the new problem of learning fair embeddings for graphs with missing attributes. Specifically, we aim to address two major challenges: (1) how to obtain meaningful node embeddings for graphs with missing attributes, and (2) how to enhance fairness of node embeddings with respect to sensitive attributes. To address these two challenges, we propose a Fair Attribute Completion (FairAC) framework. For the first challenge, we adopt an autoencoder to obtain feature embeddings for nodes with attributes and meanwhile we adopt an attention mechanism to aggregate feature information of nodes with missing attributes from their direct neighbors. Then, we address the second challenge by mitigating two types of unfairness, i.e., feature unfairness and topological unfairness. We adopt a sensitive discriminator to regulate embeddings and create a bias-free graph.

The main contributions of this paper are as follows: (1) We present a new problem of achieving fairness on a graph with missing attributes. Different from the existing work, we assume that the attributes of some nodes are entirely missing. (2) We propose a new framework, FairAC, for fair graph attribute completion, which jointly addresses unfairness issues from the feature and topology perspectives. (3) FairAC is a generic approach to complete fair graph attributes, and thus can be used in many graph-based downstream tasks. (4) Extensive experiments on benchmark datasets demonstrate the effectiveness of FairAC in eliminating unfairness and maintaining comparable accuracy.

## 2 RELATED WORK

### 2.1 FAIRNESS IN GRAPH LEARNING

Recent work promotes fairness in graph-based machine learning (Bose & Hamilton, 2019; Rahman et al., 2019; Dai & Wang, 2021; Wang et al., 2022). They can be roughly divided into three categories, i.e., the pre-processing methods, in-processing methods, and post-processing methods.

The pre-processing methods are applied before training downstream tasks by modifying training data. For instance, Fairwalk (Rahman et al., 2019) improves the sampling procedure of node2vec (Grover & Leskovec, 2016). Our FairAC framework can be viewed as a pre-processing method, as it seeks to complete node attributes and use them as input of graph neural networks. However, our problem is much harder than existing problems, because the attributes of some nodes in the graph are entirely missing, including both the sensitive ones and non-sensitive ones. Given an input graph with missing attributes, FairAC generates fair and complete feature embeddings and thus can be applied to many downstream tasks, such as node classification, link prediction (Liben-Nowell & Kleinberg, 2007; Taskar et al., 2003), PageRank (Haveliwala, 2003), etc. Graph learning models trained on the refined feature embeddings would make fair predictions in downstream tasks.

There are plenty of fair graph learning methods as in-processing solutions. Some work focus on dealing with unfairness issues on graphs with complete features. For example, GEAR (Ma et al., 2022) mitigates graph unfairness by counterfactual graph augmentation and an adversarial learning method to learn sensitive-invariant embeddings. However, in order to generate counterfactual subgraphs, they need precise and entire features for every node. In other words, it cannot work well if it encounters a graph with full missing nodes since it cannot generate counterfactual subgraph based

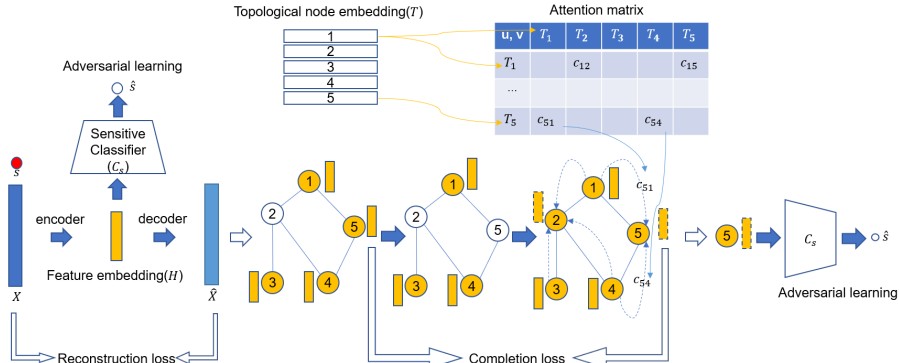

Figure 1: Overview of our FairAC framework. FairAC is composed of three major modules, i.e., an autoencoder for embedding nodes, an attributes completion module, and sensitive classifiers for mitigating feature unfairness and topological unfairness. The solid circles indicate nodes with full attributes, while the empty circles indicate nodes without any attributes.

on a blank node. But we can deal with the situation. The most related work is FairGNN (Dai & Wang, 2021). Different from the majority of problem settings on graph fairness. It learns fair GNNs for node classification in a graph where only a limited number of nodes are provided with sensitive attributes. FairGNN adopts a sensitive classifier to predict the missing sensitive labels. After that, it employs a classic adversarial model to mitigate unfairness.Specifically, a sensitive discriminator aims to predict the known or estimated sensitive attributes, while a GNN model tries to fool the sensitive discriminator and meanwhile predicts node labels. However, it cannot predict sensitive information if a node misses all features in the first place and thus will fail to achieve its final goal. Our FairAC can get rid of the problem because we recover the node embeddings from their neighbors. FairAC learns attention between neighbors according to existing full attribute nodes, so we can recover the node embeddings for missing nodes from their neighbors by aggregating the embeddings of neighbors. With the help of the adversarial learning method, it can also remove sensitive information. In addition to attribute completion, we have also designed novel de-biasing strategies to mitigate feature unfairness and topological unfairness.

## 2.2 ATTRIBUTION COMPLETION ON GRAPHS

The problem of missing attributes is ubiquitous in reality. Several methods (Liao et al., 2016; You et al., 2020; Chen et al., 2020; He et al., 2022; Jin et al., 2021; 2022; Tu et al., 2022; Taguchi et al., 2021) have been proposed to address this problem. GRAPE (You et al., 2020) tackles the problem of missing attributes in tabular data using a graph-based approach. SAT (Chen et al., 2020) assumes that the topology representation and attributes share a common latent space, and thus the missing attributes can be recovered by aligning the paired latent space. He et al. (2022) and Jin et al. (2021) extend such problem settings to heterogeneous graphs. HGNN-AC (Jin et al., 2021) is an end-to-end model, which does not recover the original attributes but generates attribute representations that have sufficient information for the final prediction task. It is worth noting that existing methods on graph attribute completion only focus on the attribute completion accuracy or performance of downstream tasks, but none of them takes fairness into consideration. Instead, our work pays attention to the unfairness issue in graph learning, and we aim to generate fair feature embeddings for each node by attribute completion, which contain the majority of information inherited from original attributes but disentangle the sensitive information.

## 3 METHODOLOGY

### 3.1 PROBLEM DEFINITION

Let $\mathcal{G} = (\mathcal{V}, \mathcal{E}, \mathcal{X})$ denote an undirected graph, where $\mathcal{V} = \{v_1, v_2, ..., v_N\}$ is the set of $N$ nodes, $\mathcal{E} \subseteq \mathcal{V} \times \mathcal{V}$ is the set of undirected edges in the graph, $\mathcal{X} \in \mathbb{R}^{N \times D}$ is the node attribute matrix, and $D$ is the dimension of attributes. $\mathcal{A} \in \mathbb{R}^{N \times N}$ is the adjacency matrix of the graph $\mathcal{G}$, where $\mathcal{A}_{ij} = 1$ if nodes $v_i$ and $v_j$ are connected; otherwise, $\mathcal{A}_{ij} = 0$. In addition, $\mathcal{S} = \{s_1, s_2, ..., s_N\}$

---

**Algorithm 1** FairAC framework algorithm

---

**Input**: $\mathcal{G} = (\mathcal{V}, \mathcal{E}, \mathcal{X}), \mathcal{S}$
**Output**: Autoencoder $f_{AE}$, Sensitive classifier $C_s$, Attribute completion $f_{AC}$

 1: Obtain topological embedding $\mathcal{T}$ with DeepWalk
 2: **repeat**
 3:     Obtain the feature embeddings H with $f_{AE}$
 4:     Optimize the $C_s$ by Equation 6
 5:     Optimize $f_{AE}$ to mitigate feature unfairness by loss $\mathcal{L}_F$
 6:     Divide $\mathcal{V}^+$ into $\mathcal{V}_{keep}$ and $\mathcal{V}_{drop}$ based on $\alpha$
 7:     Obtain the feature embeddings of nodes with missing attributes $\mathcal{V}_{drop}$ by $f_{AC}$
 8:     Optimize $f_{AC}$ to achieve attribute completion by loss $\mathcal{L}_C$
 9:     Optimize $f_{AC}$ to mitigate topological unfairness by loss $\mathcal{L}_T$
10: **until** convergence
11: **return** $f_{AE}, C_s, f_{AC}$

---

denotes a set of sensitive attributes (e.g., age or gender) of $N$ nodes, and $\mathcal{Y} = \{y_1, y_2, ..., y_N\}$ denotes the node labels. The goal of fair graph learning is to make fair predictions of node labels with respect to the sensitive attribute, which is usually measured by certain fairness notations like statistical parity (Dwork et al., 2012) and equal opportunity (Hardt et al., 2016). Statistical Parity and Equal Opportunity are two group fairness definitions. Their detailed formulations are presented below. The label $y$ denotes the ground-truth node label, and the sensitive attribute $s$ indicates one's sensitive group. For example, for binary node classification task, y only has two labels. Here we consider two sensitive groups, i.e. $s \in \{0, 1\}$.

- Statistical Parity (Dwork et al., 2012). It refers to the equal acceptance rate, which can be formulated as:
$$P(\hat{y}|s = 0) = P(\hat{y}|s = 1), \tag{1}$$
where $P(\cdot)$ denotes the probability that $\cdot$ occurs.

- Equal Opportunity (Hardt et al., 2016). It means the probability of a node in a positive class being classified as a positive outcome should be equal for both sensitive group nodes. It mathematically requires an equal true positive rate for each subgroup.
$$P(\hat{y} = 1|y = 1, s = 0) = P(\hat{y} = 1|y = 1, s = 1). \tag{2}$$

In this work, we mainly focus on addressing unfairness issues on graphs with missing attributes, i.e., attributes of some nodes are totally missing. Let $\mathcal{V}^+$ denote the set of nodes whose attributes are available, and $\mathcal{V}^-$ denote the set of nodes whose attributes are missing, $\mathcal{V} = \{\mathcal{V}^+, \mathcal{V}^-\}$. If $v_i \in \mathcal{V}^-$, both $\mathcal{X}_i$ and $s_i$ are unavailable during model training.

With the notations given below, the fair attribute completion problem is formally defined as:

**Problem 1**. *Given a graph $\mathcal{G} = (\mathcal{V}, \mathcal{E}, \mathcal{X})$, where node set $\mathcal{V}^+ \in \mathcal{V}$ with the corresponding attributes available and the corresponding sensitive attributes in $\mathcal{S}$, learn a fair attribute completion model to generate fair feature embeddings $\mathcal{H}$ for each node in $\mathcal{V}$, i.e.,*
$$f(\mathcal{G}, S) \rightarrow \mathcal{H}, \tag{3}$$
*where $f$ is the function we aim to learn. $\mathcal{H}$ should exclude any sensitive information while preserve non-sensitive information.*

## 3.2 Fair Attribute Completion (FairAC) Framework

We propose a fair attribute completion (FairAC) framework to address Problem 1. Existing fair graph learning methods tackle unfairness issues by training fair graph neural networks in an end-to-end fashion, but they cannot effectively handle graphs that are severely biased due to missing attributes. Our FairAC framework, as a data-centric approach, deals with the unfairness issue from a new perspective, by explicitly debiasing the graph with feature unfairness mitigation and fairness-aware attribute completion. Eventually, FairAC generates fair embeddings for all nodes including the ones without any attributes. The training algorithms are shown in Algorithm 1.

To train the graph attribute completion model, we follow the setting in (Jin et al., 2021) and divide the nodes with attributes (i.e., $\mathcal{V}^+$) into two sets: $\mathcal{V}_{keep}$ and $\mathcal{V}_{drop}$. For nodes in $\mathcal{V}_{keep}$, we keep their attributes, while for nodes in $\mathcal{V}_{drop}$, we temporally drop their attributes and try to recover them using our attribute completion model. Although the nodes are randomly assigned to $\mathcal{V}_{keep}$ and $\mathcal{V}_{drop}$, the proportion of $\mathcal{V}_{drop}$ is consistent with the attribute missing rate $\alpha$ of graph $\mathcal{G}$, i.e., $\alpha = \frac{|\mathcal{V}^-|}{|\mathcal{V}|} = \frac{|\mathcal{V}_{drop}|}{|\mathcal{V}^+|}$.

Different from existing work on fair graph learning, we consider unfairness from two sources. The first one is from node features. For example, we can roughly infer one's sensitive information, like gender, from some non-sensitive attributes like hobbies. It means that non-sensitive attributes may imply sensitive attributes and thus lead to unfairness in model prediction. We adopt a sensitive discriminator to mitigate feature unfairness. The other source is topological unfairness introduced by graph topological embeddings and node attribute completion. To deal with the topological unfairness, we force the estimated feature embeddings to fool the sensitive discriminator, by updating attention parameters during the attribute completion process.

As illustrated in Figure 1, our FairAC framework first mitigates feature unfairness for nodes with attributes (i.e., $\mathcal{V}_{keep}$) by removing sensitive information implicitly contained in non-sensitive attributes with an auto-encoder and sensitive classifier (Section 3.2.1). For nodes without features (i.e., $\mathcal{V}_{drop}$), FairAC performs attribute completion with an attention mechanism (Section 3.2.2) and meanwhile mitigates the topological unfairness (Section 3.2.3). Finally, the FairAC model trained on $\mathcal{V}_{keep}$ and $\mathcal{V}_{drop}$ can be used to infer fair embeddings for nodes in $\mathcal{V}^-$.

The overall loss function of FairAC is formulated as:

$$\mathcal{L} = \mathcal{L}_F + \mathcal{L}_C + \beta\mathcal{L}_T, \tag{4}$$

where $\mathcal{L}_F$ represents the loss for mitigating feature unfairness, $\mathcal{L}_C$ is the loss for attribute completion, and $\mathcal{L}_T$ is the loss for mitigating topological unfairness. $\beta$ is a trade-off hyperparameter.

### 3.2.1 MITIGATING FEATURE UNFAIRNESS

The nodes in $\mathcal{V}_{keep}$ have full attributes $\mathcal{X}$, while some attributes may implicitly encode information about sensitive attributes $\mathcal{S}$ and thus lead to unfair predictions. To address this issue, FairAC aims to encode the attributes $\mathcal{X}_i$ of node $i$ into a fair feature embedding $\mathcal{H}_i$. Specifically, we use a simple autoencoder framework together with a sensitive classifier. The autoencoder maps $\mathcal{X}_i$ into embedding $\mathcal{H}_i$, and meanwhile the sensitive classifier $C_s$ is trained in an adversarial way, such that the embeddings are invariant to sensitive attributes.

**Autoencoder.** The autoencoder contains an encoder $f_E$ and a decoder $f_D$. $f_E$ encodes the original attributes $\mathcal{X}_i$ to feature embeddings $\mathcal{H}_i$, i.e., $\mathcal{H}_i = f_E(\mathcal{X}_i)$, and $f_D$ reconstructs attributes from the latent embeddings, i.e., $\hat{\mathcal{X}}_i = f_D(\mathcal{H}_i)$, where the reconstructed attributes $\hat{\mathcal{X}}$ should be close to $\mathcal{X}_i$ as possible. The loss function of the autoencoder is written as:

$$\mathcal{L}_{ae} = \frac{1}{|\mathcal{V}_{keep}|} \sum_{i \in \mathcal{V}_{keep|}} \sqrt{(\hat{\mathcal{X}}_i - \mathcal{X}_i)^2}. \tag{5}$$

**Sensitive classifier** The sensitive classifier $C_s$ is a simple multilayer perceptron (MLP) model. It takes the feature embedding $\mathcal{H}_i$ as input and predicts the sensitive attribute $\hat{s}_i$, i.e., $\hat{s}_i = C_s(\mathcal{H}_i)$. When the sensitive attributes are binary, we can use the binary cross entropy loss to optimize $C_s$:

$$\mathcal{L}_{C_s} = -\frac{1}{|\mathcal{V}_{keep}|} \sum_{i \in \mathcal{V}_{keep}} s_i \log \hat{s}_i + (1 - s_i) \log (1 - \hat{s}_i). \tag{6}$$

With the sensitive classifier $C_s$, we could leverage it to adversarially train the autoencoder, such that $f_E$ is able to generate fair feature embeddings that can fool $C_s$. The loss $\mathcal{L}_F$ is written as: $\mathcal{L}_F = \mathcal{L}_{ae} - \beta\mathcal{L}_{C_s}$.

### 3.2.2 COMPLETING NODE EMBEDDINGS VIA ATTENTION MECHANISM

For nodes without attributes ($\mathcal{V}_{drop}$), FairAC makes use of topological embeddings and completes the node embeddings $\mathcal{H}_{drop}$ with an attention mechanism.

**Topological embeddings.**   Recent studies reveal that the topology of graphs has similar semantic information as the attributes (Chen et al., 2020; McPherson et al., 2001; Pei et al., 2020; Zhu et al., 2020). Inspired by this observation, we assume that the nodes' topological information can reflect the relationship between nodes' attributes and the attributes of their neighbors. There are a lot of off-the-shelf node topological embedding methods, such as DeepWalk (Perozzi et al., 2014) and node2vec (Grover & Leskovec, 2016). For simplicity, we adopt the DeepWalk method to extract topological embeddings for nodes in $\mathcal{V}$.

**Attention mechanism.**   For graphs with missing attributes, a commonly used strategy is to use average attributes of the one-hop neighbors. This strategy works in some cases, however, simply averaging information from neighbors might be biased, as the results might be dominated by some high-degree nodes. In fact, different neighbors should have varying contributions to the aggregation process in the context of fairness. To this end, FairAC adopts an attention mechanism (Vaswani et al., 2017) to learn the influence of different neighbors or edges with the awareness of fairness, and then aggregates attributes information for nodes in $\mathcal{V}_{drop}$.

Given a pair of nodes $(u, v)$ which are neighbors, the contribution of node $v$ is the attention $att_{u,v}$, which is defined as: $att_{u,v} = Attention(T_u, T_v)$, where $T_u, T_v$ are the topological embeddings of nodes $u$ and $v$, respectively. Specifically, we only focus on the neighbor pairs and ignore those node pairs that are not directly connected. $Attention(\cdot, \cdot)$ denotes the attention between two topological embeddings, i.e., $Attention(T_u, T_v) = \sigma(T_u^T W T_v)$, where $W$ is the learnable parametric matrix, and $\sigma$ is an activation function. After we get all the attention scores between one node and its neighbors, we can get the coefficient of each pair by applying the softmax function:

$$c_{u,v} = \text{softmax}(att_{u,v}) = \frac{\exp(att_{u,v})}{\sum_{s \in N_u} \exp(att_{u,s})}, \tag{7}$$

where $c_{u,v}$ is the coefficient of node pair $(u, v)$, and $N_u$ is the set of neighbors of node $u$. For node $u$, FairAC calculates its feature embedding $\hat{\mathcal{H}}_u$ by the weighted aggregation with multi-head attention:

$$\hat{\mathcal{H}}_u = \frac{1}{K} \sum_{k=1}^{K} \sum_{s \in N_u} c_{u,s} \mathcal{H}_s, \tag{8}$$

where $K$ is the number of attention heads. The loss for attribute completion with topological embedding and attention mechanism is formulated as:

$$\mathcal{L}_C = \frac{1}{|\mathcal{V}_{drop}|} \sum_{i \in \mathcal{V}_{drop}|} \sqrt{(\hat{\mathcal{H}}_i - \mathcal{H}_i)^2}. \tag{9}$$

### 3.2.3   MITIGATING TOPOLOGICAL UNFAIRNESS

The attribute completion procedure may introduce topological unfairness since we assume that topology information is similar to attributes relation. It is possible that the completed feature embeddings of $\mathcal{V}_{drop}$ would be unfair with respect to sensitive attributes $\mathcal{S}$. To address this issue, FairAC leverages sensitive classifier $C_s$ to help mitigate topological unfairness by further updating the attention parameter matrix $W$ and thus obtaining fair feature embeddings $\mathcal{H}$. Inspired by (Gong et al., 2020), we expect that the feature embeddings can fool the sensitive classifier $C_s$ to predict the probability distribution close to the uniform distribution over the sensitive category, by minimizing the loss:

$$\mathcal{L}_T = -\frac{1}{|\mathcal{V}_{drop}|} \sum_{i \in \mathcal{V}_{drop}} s_i \log \hat{s}_i + (1 - s_i) \log (1 - \hat{s}_i). \tag{10}$$

### 3.3   FAIRAC FOR NODE CLASSIFICATION

The proposed FairAC framework could be viewed as a generic data debiasing approach, which achieves fairness-aware attribute completion and node embedding for graphs with missing attributes. It can be easily integrated with many existing graph neural networks (e.g., GCN (Kipf & Welling, 2016), GAT (Veličković et al., 2018), and GraphSAGE (Hamilton et al., 2017)) for tasks like node classification. In this work, we choose the basic GCN model for node classification and assess how FairAC enhances model performance in terms of accuracy and fairness.

Table 1: Comparisons of our FairAC method and baselines on three graphs. M refers to missing or not. M is true means that some nodes' attributes are entirely missing and the ratio is controlled by $\alpha$. Otherwise, full attributes are provided. The attribute missing rate $\alpha$ is set to 0.3. GCN and FairGNN are trained on averaging attribute completed graphs. Bold fonts denote the best results.

| Dataset | Method | M | Acc ↑ | AUC ↑ | $\Delta SP$ ↓ | $\Delta EO$ ↓ | $\Delta SP+\Delta EO$ ↓ |
|---------|--------|---|-------|-------|------|------|------------|
| NBA | GCN | ✓ | **70.66±0.24** | 74.23±0.63 | 3.43±2.44 | 2.74±0.67 | 6.16±3.1 |
| | ALFR | × | 64.3±1.3 | 71.5±0.3 | 2.3±0.9 | 3.2±1.5 | 5.5±2.4 |
| | ALFR-e | × | 66.0±0.4 | 72.9±1.0 | 4.7±1.8 | 4.7±1.7 | 9.4±3.4 |
| | Debias | × | 63.1±1.1 | 71.3±0.7 | 2.5±1.5 | 3.1±1.9 | 5.6±3.4 |
| | Debias-e | × | 65.6±2.4 | 72.9±1.2 | 5.3±0.9 | 3.1±1.3 | 8.4±2.2 |
| | FCGE | × | 66.0±1.5 | 73.6±1.5 | 2.9±1.0 | 3.0±1.2 | 5.9±2.2 |
| | FairGNN | ✓ | 70.73±0.44 | **76.77±0.1** | 0.95±0.7 | 1.63±0.67 | 2.58±1.37 |
| | FairAC (Ours) | ✓ | **70.66±0.73** | 74.44±0.67 | **0.28±0.25** | **0.63±0.34** | **0.91±0.59** |
| Pokec-z | GCN | ✓ | 67.4±0.88 | 72.04±1.78 | 1.96±0.64 | 4.17±0.54 | 6.12±1.18 |
| | ALFR | × | 65.4±0.3 | 71.3±0.3 | 2.8±0.5 | 1.1±0.4 | 3.9±0.9 |
| | ALFR-e | × | **68.0±0.6** | **74.0±0.7** | 5.8±0.4 | 2.8±0.8 | 8.6±1.2 |
| | Debias | × | 65.2±0.7 | 71.4±0.6 | 1.9±0.6 | 1.9±0.4 | 3.8±1.0 |
| | Debias-e | × | 67.5±0.7 | 74.2±0.7 | 4.7±1.0 | 3.0±1.4 | 7.7±2.4 |
| | FCGE | × | 65.9±0.2 | 71.0±0.2 | 3.1±0.5 | 1.7±0.6 | 4.8±1.1 |
| | FairGNN | ✓ | 66.54±0.45 | 70.10±0.07 | 0.95±0.70 | 1.63±0.67 | 2.58±0.57 |
| | FairAC (Ours) | ✓ | 66.94±0.14 | 72.87±0.13 | **0.19±0.07** | **0.12±0.07** | **0.31±0.14** |
| Pokec-n | GCN | ✓ | 66.12±0.88 | 71.5±0.1 | 0.46±0.1 | 1.41±0.14 | 1.87±0.24 |
| | ALFR | × | 63.1±0.6 | 67.7±0.5 | 3.05±0.5 | 3.9±0.6 | 3.95±1.1 |
| | ALFR-e | × | 66.2±0.4 | 71.9±1.0 | 4.1±1.8 | 4.6±1.7 | 8.7±3.5 |
| | Debias | × | 62.6±1.1 | 67.9±0.7 | 2.4±1.5 | 2.6±1.9 | 5.0±3.4 |
| | Debias-e | × | 65.6±2.4 | 71.7±1.2 | 3.6±0.9 | 4.4±1.3 | 8.0±2.2 |
| | FCGE | × | 64.8±1.5 | 69.5±1.5 | 4.1±1.0 | 5.5±1.2 | 9.6±2.2 |
| | FairGNN | ✓ | **68.54±0.45** | 70.10±0.07 | 0.76±0.15 | 0.48±0.09 | 1.24±0.24 |
| | FairAC (Ours) | ✓ | 66.35±0.24 | **72.32±0.08** | **0.27±0.12** | **0.14±0.11** | **0.41±0.23** |

## 4 EXPERIMENTS

In this section, we evaluate the performance of the proposed FairAC framework on three benchmark datasets in terms of node classification accuracy and fairness w.r.t. sensitive attributes. We compare FairAC with other baseline methods in settings with various sensitive attributes or different attribute missing rates. Ablation studies are also provided and discussed.

### 4.1 DATASETS AND SETTINGS

**Datasets.** In the experiments, we use three public graph datasets, **NBA**, **Pokec-z**, and **Pokec-n**. A detailed description is shown in supplementary materials.

**Baselines.** We compare our FairAC method with the following baseline methods: GCN (Kipf & Welling, 2016), ALFR (Edwards & Storkey, 2015), ALFR-e, Debias (Zhang et al., 2018), Debias-e, FCGE (Bose & Hamilton, 2019), and FairGNN (Dai & Wang, 2021). ALFR-e concatenates the feature embeddings produced by ALFR with topological embeddings learned by DeepWalk (Perozzi et al., 2014). Debias-e also concatenates the topological embeddings learned by DeepWalk with feature embeddings learned by Debias. FairGNN is an end-to-end debias method which aims to mitigate unfairness in label prediction task. GCN and FairGNN uses the average attribute completion method, while other baselines use original complete attributes.

**Evaluation Metrics.** We evaluate the proposed framework with respect to two aspects: classification performance and fairness performance. For classification, we use accuracy and AUC scores. As for fairness, we adopt $\Delta SP$ and $\Delta EO$ as evaluation metrics, which can be defined as:

$$\Delta SP = P(\hat{y}|s=0) - P(\hat{y}|s=1), \tag{11}$$

$$\Delta EO = P(\hat{y}=1|y=1,s=0) - P(\hat{y}=1|y=1,s=1). \tag{12}$$

The smaller $\Delta SP$ and $\Delta EO$ are, the more fair the model is. In addition, we use $\Delta SP+\Delta EO$ as an overall indicator of a model's performance on fairness.

Table 2: Comparisons of our method with the baselines on pokec-z dataset with four levels of attribute missing rates $\alpha$. FairAC generates fair and complete node features, and then GCN is trained for node classification. BaseAC is a simplified version of FairAC, which only has the attention-based attribute completion module, but does not contain the module for mitigating feature unfairness and topological unfairness. Bold fonts denote the best results.

| $\alpha$ | Method | Acc (%)↑ | AUC (%)↑ | $\Delta SP$ (%)↓ | $\Delta EO$ (%)↓ | $\Delta SP+\Delta EO$ ↓ |
|---|---|---|---|---|---|---|
| | GCN | 66.10 | 69.14 | 0.88 | 0.20 | 1.08 |
| 0.1 | FairGNN | **69.37** | **76.93** | 0.17 | **0.29** | 0.46 |
| | BaseAC (Ours) | 66.37 | 69.34 | **0.07** | 1.46 | 1.53 |
| | FairAC (Ours) | 66.33 | 69.35 | 0.14 | 0.37 | **0.51** |
| | GCN | 67.40 | 72.04 | 1.96 | 4.17 | 6.12 |
| 0.3 | FairGNN | 66.54 | 70.10 | 0.95 | 1.63 | 2.58 |
| | BaseAC (Ours) | 66.10 | 69.67 | 0.78 | 1.57 | 2.35 |
| | FairAC (Ours) | **66.94** | **72.87** | **0.19** | **0.12** | **0.31** |
| | GCN | 66.41 | 70.14 | 0.07 | 2.11 | 2.18 |
| 0.5 | FairGNN | 66.25 | 70.13 | 0.47 | 1.71 | 2.18 |
| | BaseAC (Ours) | 66.13 | 69.93 | 0.38 | 1.60 | 1.98 |
| | FairAC (Ours) | **66.45** | **72.95** | **0.14** | **1.06** | **1.20** |
| | GCN | **66.99** | **73.13** | 0.57 | 1.74 | 2.31 |
| 0.8 | FairGNN | 66.60 | 71.35 | 0.94 | **0.09** | 1.03 |
| | BaseAC (Ours) | 66.06 | 71.21 | 0.08 | 2.20 | 2.28 |
| | FairAC (Ours) | 66.10 | 71.66 | **0.01** | 0.69 | **0.70** |

## 4.2 Results and Analysis

### 4.2.1 Unfairness issues in Graph Neural Networks

According to the results showed in Table 1, they reveal several unfairness issues in Graph Neural Networks. We divided them into two categories.

- **Feature unfairness** Feature unfairness is that some non-sensitive attributes could infer sensitive information. Hence, some Graph Neural Networks may learn this relation and make unfair prediction. In most cases, ALFR and Debias and FCGE have better fairness performance than GCN method. It is as expected because the non-sensitive features may contain proxy variables of sensitive attributes which would lead to biased prediction. Thus, ALFR and Debias methods that try to break up these connections are able to mitigate feature unfairness and obtain better fairness performance. These results further prove the existence of feature unfairness.

- **Topological unfairness** Topological unfairness is sourced from graph structure. In other words, edges in graph, i.e. the misrepresentation due to the connection(Mehrabi et al., 2021) can bring topological unfairness. From the experiments, ALFR-e and Debias-e have worse fairness performance than ALFR and Debias, respectively. It shows that although graph structure can improve the classification performance, it will bring topological unfairness consequently. The worse performance on fairness verifies that topological unfairness exists in GNNs and graph topological information could magnify the discrimination.

### 4.2.2 Effectiveness of FairAC on mitigating feature and topological unfairness

The results of our FairAC method and baselines in terms of the node classification accuracy and fairness metrics on three datasets are shown in Table 1. The best results are shown in bold. Generally speaking, we have the following observations. (1). The proposed method FairAC shows comparable classification performance with these baselines, GCN and FairGNN. This suggests that our attribute completion method is able to preserve useful information contained in the original attributes. (2).

FairAC outperforms all baselines regarding fairness metrics, especially in $\Delta SP + \Delta EO$. FairAC outperform baselines that focus on mitigate feature fairness, like ALFR, which proves that FairAC also mitigate topological unfairness. Besides, it is better than those who take topological fairness into consideration, like FCGE, which also validates the effectiveness of FairAC. FairGNN also has good performance on fairness, because it adopts a discriminator to deal with the unfairness issue. Our method performs better than FairGNN in most cases. For example, our FairAC method can significantly improve the performance in terms of the fairness metric $\Delta SP + \Delta EO$, i.e., 65%, 87%, and 67% improvement over FairGNN on the NBA, pokec-z, pokec-n datasets, respectively. Overall, the results in Table 1 validate the effectiveness of FairAC in mitigating unfairness issues.

## 4.3 ABLATION STUDIES

**Attribute missing rate** In our proposed framework, the attribute missing rate indicates the integrity of node attribute matrix, which has a great impact on model performance. Here we investigate the performance of our FairAC method and baselines on dealing with graphs with varying degrees of missing attributes. In particular, we set the attribute missing rate to 0.1, 0.3, 0.5 and 0.8, and evaluate FairAC and baselines on the pokec-z dataset. The detailed results are presented in Table 2. From the table, we have the following observation that with varying values of $\alpha$, FairAC is able to maintain its high fairness performance. Especially when $\alpha$ reaches 0.8, FairAC can greatly outperform other methods. It proves that FairAC is effective even if the attributes are largely missing.

**The effectiveness of adversarial learning** A key module in FairAC is adversarial learning, which is used to mitigate feature unfairness and topological unfairness. To investigate the contribution of adversarial learning in FairAC, we implement a BaseAC model, which only has the attention-based attribute completion module, but does not contain the adversarial learning loss terms. Comparing BaseAC with FairAC in Table 2, we can find that the fairness performance drops desperately when the adversarial training loss is removed. Since BaseAC does not have an adversarial discriminator to regulate feature encoder as well as attribute completion parameters, it is unable to mitigate unfairness. Overall, the results confirm the effectiveness of the adversarial learning module.

**Parameter analysis** We investigate how the hyperparameters affect the performance of FairAC. The most important hyperparameter in FairAC is $\beta$, which adjusts the trade-off between fairness and attribute completion. We report the results with different hyperparameter values. We set $\beta$ to 0.2, 0.4, 0.7, 0.8 and 0 that is equivalent to the BaseAC. We also fix other hyperparameters by setting $\alpha$ to 0.3. As shown in Figure 2, we can find that, as $\beta$ increases, the fairness performance improves while the accuracy of node classification slightly declined. Therefore, it validates our assumption that there is a trade-off between fairness and attribute completion, and our FairAC is able to enhance fairness without compromising too much on accuracy.

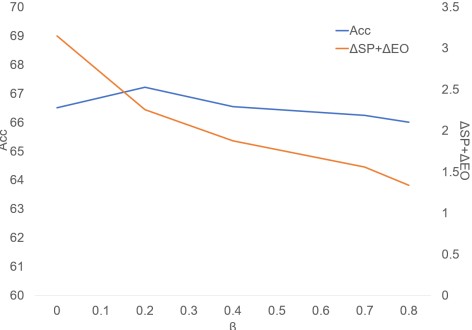

Figure 2: Accuracy and $\Delta SP + \Delta EO$ of FairAC when varying $\beta$ on Pokec-z dataset with $\alpha = 0.3$.

## 5 CONCLUSIONS

In this paper, we presented a novel problem, i.e., fair attribute completion on graphs with missing attributes. To address this problem, we proposed the FairAC framework, which jointly completes the missing features and mitigates unfairness. FairAC leverages the attention mechanism to complete missing attributes and adopts a sensitive classifier to mitigate implicit feature unfairness as well as topological unfairness on graphs. Experimental results on three real-world datasets demonstrate the superiority of the proposed FairAC framework over baselines in terms of both node classification performance and fairness performance. As a generic fair graph attributes completion approach, FairAC can also be used in other graph-based downstream tasks, such as link prediction, graph regression, pagerank, and clustering.

ACKNOWLEDGEMENT

This research is supported by the Cisco Faculty Award and Adobe Data Science Research Award.

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

Table 3: Statistics of three graph datasets.

| Dataset | NBA | Pokec-z | Pokec-n |
|---------|-----|---------|---------|
| # of nodes | 403 | 67,797 | 66,569 |
| # of edges | 16,570 | 882,765 | 729,129 |
| Density | 0.10228 | 0.00019 | 0.00016 |

## A APPENDIX

### A.1 DATASETS AND SETTINGS

**Datasets.** In the experiments, we use three public graph datasets, **NBA**, **Pokec-z**, and **Pokec-n**. The detailed explanation is shown in supplementary materials. The NBA dataset (Dai & Wang, 2021) is extended from a Kaggle dataset containing around 400 NBA basketball players. It provides the performance statistics of those players in the 2016-2017 season and their personal profiles, e.g., nationality, age, and salary. Their relationships are obtained from Twitter. We use their nationality, whether one is U.S. player or oversea player, as the sensitive attribute. The node label is binary, indicating whether the salary of the player is over median or not. Pokec (Takac & Zabovsky, 2012) is an online social network in Slovakia, which contains millions of anonymized data of users. It has a variety of attributes, such as gender, age, education, region, etc. Based on the region where users belong to, (Dai & Wang, 2021) sampled two datasets named as: Pokec-z and Pokec-n. In our experiments, we consider the region or gender as sensitive attribute, and working field as label for node classification. The statistics of three datasets are summarized in supplementary materials. The statistics of three datasets are summarized in Table 3.

**Baselines.** We compare our FairAC method with the following baseline methods:

- **GCN (Kipf & Welling, 2016) with average attribute completion.** GCN is a classical graph neural network model, which has obtained very promising performance in numerous applications. The standard GCN cannot handle graphs with missing attributes. In the experiments, we use the average attribute completion strategy to preprocess the feature matrix, by using the averaged attributes of one's neighbors to approximate the missing attributes. After average attribute completion, GCN takes the graph with completed feature matrix as inputs to learn node embeddings and predict node labels.

- **ALFR (Edwards & Storkey, 2015) with full attributes.** This is a pre-processing method. It utilize a discriminator to remove the sensitive feature information in feature embeddings produced by an Autoencoder. Since this method need full sensitive attributes and full features, we give them complete information. In other words, the missing rate $\alpha$ is set to 0.

- **ALFR-e with full attributes.** Based on ALFR, ALFR-e utilize the topological information. It concatenates the feature embeddings produced by ALFR with topological embeddings learned by DeepWalk (Perozzi et al., 2014). It also relys on complete information.

- **Debias (Zhang et al., 2018) with full attributes.** This is an in-processing method. It applies a discriminator on node classifier in order to make the probability distribution be the same w.r.t. sensitive attribute. Since the discriminator needs the full sensitive attributes, we provide full node features.

- **Debias-e with full attributes.** Similar to ALFR-e. It also concatenates the topological embeddings learned by DeepWalk (Perozzi et al., 2014) with feature embeddings learned by Debias.

- **FCGE (Bose & Hamilton, 2019) with full attributes.** It learns fair node embeddings in graph without node features through edge prediction only. An discriminator is also applied to mitigate sensitive information in topological perspective.

- **FairGNN (Dai & Wang, 2021) with average attribute completion.** Although FairGNN trains a sensitive attribute discriminator as an adversarial regularizer to enhance the fairness

Table 4: Comparisons of our method with the baselines on pokec-n dataset with three levels of attribute missing rates $\alpha$. FairAC generates fair and complete node features, and then GAT is trained for node classification. Bold fonts denote the best results.

| $\alpha$ | Method | Acc (%)↑ | AUC (%)↑ | $\Delta SP$ (%)↓ | $\Delta EO$ (%)↓ | $\Delta SP + \Delta EO$ ↓ |
|---|---|---|---|---|---|---|
| 0.3 | GAT | **67.77** | **73.57** | 1.02 | 3.38 | 3.40 |
| | FairGNN | 66.55 | 68.64 | 0.45 | 0.99 | 1.44 |
| | FairAC (Ours) | 67.25 | 72.96 | **0.23** | **0.10** | **0.33** |
| 0.5 | GAT | **68.59** | **73.97** | 0.30 | 1.96 | 2.26 |
| | FairGNN | 68.09 | 72.22 | 0.81 | 1.55 | 2.36 |
| | FairAC (Ours) | 66.36 | 70.66 | **0.09** | **0.32** | **0.41** |
| 0.7 | GAT | **67.59** | **71.62** | 3.69 | 7.15 | 10.84 |
| | FairGNN | 62.36 | 67.99 | 2.95 | **3.55** | 6.50 |
| | FairAC (Ours) | 66.64 | 69.59 | **0.18** | 4.19 | **4.37** |

of GNNs, it still cannot deal with graphs with missing attributes. Thus, we use the average attribute completion method to complete the feature matrix, and then train a FairGNN model for node classification.

**Implementation Details.** Each dataset is randomly split into 75%/25% training/test set as (Dai & Wang, 2021). Besides, we randomly drop node attributes based on the attribute missing rate, $\alpha$, which means the attributes of $\alpha \times |\mathcal{V}|$ nodes will be unavailable. For each datasets, we choose a specific attribute as the sensitive attribute. In particular, *region*, and *nation* are selected as the sensitive attribute for the pokec, and nba datasets, respectively. Unless otherwise specified, we generate 128-dimension node embeddings and set the attribute missing rate $\alpha$ to 0.3, and set the hyperparameters of FairAC as: $\beta = 1$ for pokec-z and nba datasets, and $\beta = 0.5$ for pokec-n dataset. We adopt Adam (Kingma & Ba, 2014) with the learning rate of 0.001 and weight decay as $1e - 5$. We adopt the DeepWalk (Perozzi et al., 2014) method to generate topological embedding for each node. Specifically, we use the DeepWalk implementation provided by the Karate Club library (Rozemberczki et al., 2020). We set walk length as 100, embedding dimension as 64, window size as 5, and epochs as 10. To evaluate fairness of compared methods, we follow the widely used evaluation protocol in fair graph learning and set a threshold for accuracy, because there is a trade-off between accuracy and fairness. Since we mainly focus on the fairness metric, we set the accuracy threshold that all methods can satisfy. we evaluated our models three times and calculated the mean and standard deviation(std). We estimate the std of $\Delta SP + \Delta EO$ by adding std of $\Delta SP$ and $\Delta EO$, because for some methods, we use the reported data from (Dai & Wang, 2021) which does not provide the metric.

## A.2 ADDITIONAL EXPERIMENTS

**Evaluations on GAT (Veličković et al., 2018) model.** As discussed in the main paper, the proposed FairAC method can be easily integrated with existing graph neural networks. Extensive results in Section 4 of the main paper demonstrate that the combination of FairAC and GCN performs very well. In this section, we integrate FairAC with another representative graph neural network model, GAT (Veličković et al., 2018). The results of our method and two main baselines in terms of the node classification accuracy and fairness metrics are shown in Table 4. In these experiments, FairAC generates fair and complete node features, and then GAT is trained for node classification. We also investigate the performance of our FairAC method and baselines on dealing with graphs with varying degrees of missing attributes. We set the attribute missing rate to 0.1, 0.3, 0.5 and 0.7, and evaluate FairAC and baselines on the Pokec-n dataset. In addition, we set $\beta$ to 1.0. The best results are shown in bold. Generally speaking, we have the following observations. (1). The proposed method FairAC shows comparable classification performance with two baselines, GAT and FairGNN. This suggests that our attribute completion method is able to work well under different downstream models. It further demonstrates that FairAC can preserve useful information implied in the original attributes. (2). FairAC has comparable results with two baselines regarding fairness

metrics. Especially when $\alpha$ is greater than 0.3, FairAC can greatly outperform other methods, which proves that FairAC is effective even if the attributes are largely missing. Overall, the results in Table 4 validate the effectiveness of FairAC in mitigating unfairness issues and show the compatibility with varying downstream models.

