# OpenReview forum: "Fair Attribute Completion on Graph with Missing Attributes"
_ICLR.cc/2023/Conference — ICLR 2023 poster_

### Official Review · Reviewer_Rh4z · 2022-10-24

**Confidence:** 4
**Correctness:** 3
**Technical Novelty And Significance:** 3
**Empirical Novelty And Significance:** 3
**Recommendation:** 6

**Clarity, Quality, Novelty And Reproducibility:**

Clarity:Good: The paper is well organized, but the presentation has minor details that could be improved.

Quality:Good: The paper appears to be technically sound. The proofs, if applicable, appear to be correct, but I have not carefully checked the details. The experimental evaluation, if applicable, is adequate, and the results convincingly support the main claims.

Novelty:Fair: The paper contributes some new ideas or represents incremental advances.

Reproducibility: Good: key resources (e.g., proofs, code, data) are available.

**Strength And Weaknesses:**

Strengths

1.	The topic is interesting and important for the graph representation learning community.

2.	This work introduces a valuable learning problem, i.e., fairness on a graph with missing attributes. The paper is well-organized and easy to follow.

3.	The idea is new, and the proposed algorithm is rational.

Weaknesses:

1. Some works [1-4] are not addressed in the related work section nor in the experimental evaluation. In my opinion, the authors should compare their solution with these algorithms at least in the related work section.

[1] Amer: A New Attribute-Missing Network Embedding Approach. TCYB, 2022.

[2] Initializing Then Refining: A Simple Graph Attribute Imputation Network. IJCAI, 2022.

[3] Graph Convolutional Networks for Graphs Containing Missing Features. FGCS, 2021.

[4] Heterogeneous Graph Neural Network via Attribute Completion. WWW, 2021.

2. Although this paper is well-motivated, the technique contributions seem a bit weak. The used techniques such as attention mechanism, adversarial learning, and auto-encoder have been widely researched in graph domains. I wonder the reason that why combining these components contributes more to the performance.

3.  According to the results from ablation studies, the effectiveness of the proposed components seems unclear. For example, Table 2 can see no difference in the performance of some datasets.

4. Explain if the metric is for one experiment over randomized trials, e.g., randomized initialization. The results in Table 1 and Table 2 would be stronger if the metric reported were mean/std.

5. I would discuss the difference and the connection between the proposed model and existing graph fairness methods in theory.


**Summary Of The Paper:**

In this paper, the authors study an important graph embedding issue, i.e., fairness on attribute-missing graphs. They propose a new fair attribute completion method to solve it by considering both feature and topological unfairness. The experiments show comparable results of the proposed method on some benchmarks.

**Summary Of The Review:**

The paper has studied a very interesting problem. Besides, the idea of the paper is novel and the work is technique solid.

---

> ### Author Response · Authors · 2022-11-17
> **Thank you for your valuable comments and feedback. We would like to address your concerns and questions.**
>
>
> We have uploaded a new version of the paper with changes based on your feedback, for which we are thankful.
>
> ### Q1
>
>
>
> Thank you very much for suggesting the related work. We have added these references and discussed them in the related work section.
>
>
>
> ### Q2
>
> >Although this paper is well-motivated, the technique contributions seem a bit weak. The used techniques such as attention mechanism, adversarial learning, and auto-encoder have been widely researched in graph domains. I wonder the reason that why combining these components contributes more to the performance.
>
>
>
> The main contributions of our work include a new problem setting on fair graph learning, as well as a simple yet effective solution to it.
>
> 1.  Auto-encoder and attention mechanism have been validated as very fundamental and effective designs in various deep learning problems, including deep graph learning. We adopt them to build an extensible graph learning framework, such that our framework can be easily extended to other graph-based downstream applications.
>
> 2.  Adversarial learning was used in our framework, as it perfectly matches our objective, i.e., learning embeddings that are invariant to sensitive attributes. We have also evaluated the effectiveness of adversarial learning in experiments.
>
>
> In summary, the adversarial learning component is a key component in our framework towards the fair learning objective, while auto-encoder and attention could be possibly replaced by other advanced designs.
>
>
>
>
>
> ### Q3
>
> >According to the results from ablation studies, the effectiveness of the proposed components seems unclear. For example, Table 2 can see no difference in the performance of some datasets.
>
>
>
> To evaluate fairness of compared methods, we follow the widely used evaluation protocol in fair graph learning and set a threshold for accuracy, because there is a trade-off between accuracy and fairness. Since we mainly focus on the fairness metric, we set the accuracy threshold that all methods can satisfy, e.g., around 66% on the pokec_n dataset. In Table 1 and Table 2, our method obtains the best fairness performance and can maintain a comparable performance on accuracy and AUC.
>
>
>
>
> ### Q4
>
> >Explain if the metric is for one experiment over randomized trials, e.g., randomized initialization. The results in Table 1 and Table 2 would be stronger if the metric reported were mean/std.
>
>
>
> The results reported in our paper are average results. In the revised paper, we have also added the standard deviations (std) of results in Table 1. Specifically, we evaluated our models three times and calculated the mean and std.
>
>
>
> ### Q5
>
> >I would discuss the difference and the connection between the proposed model and existing graph fairness methods in theory.
>
>
>
> Thanks again for suggesting related work on graph fairness. The most related work is FairGNN [2]. It learns fair GNNs for node classification in a graph where only a limited number of nodes are provided with sensitive attributes. FairGNN adopts a sensitive classifier to predict the missing sensitive labels. After that, It employs a classic adversarial model to mitigate unfairness.Specifically, a sensitive discriminator aims to predict the known or estimated sensitive attributes, while a GNN model tries to fool the sensitive discriminator and meanwhile predicts node labels. However, it cannot predict sensitive information if a node misses all the features in the first place and thus will fail to achieve their final goal. Our FairAC can get rid of the problem because we recover the node embeddings from their neighbors. FairAC learns attention between neighbors according to existing full attribute nodes, so we can recover the node embeddings for missing nodes from their neighbors by aggregating neighbors’ embeddings. With the help of adversarial learning method, it can also remove sensitive information.
>
>
>
> Other works focus on dealing with unfairness issues on graphs with complete features. For example, GEAR [1] mitigates graph unfairness by counterfactual graph augmentation and an adversarial learning method to learn sensitive-invariant embeddings. However, in order to generate counterfactual subgraph, they need precise and entire features for every node. In other words, It cannot work well if it encounters a graph with full missing nodes since it cannot generate counterfactual subgraph based on a blank node. But we can deal with the situation.
>
>
>
> [1] Ma, Jing, et al. "Learning fair node representations with graph counterfactual fairness." WSDM. 2022.
>
> [2] Enyan Dai and Suhang Wang. Say no to the discrimination: Learning fair graph neural networks with limited sensitive attribute information.WSDM, 2021.

---

> ### Author Response · Authors · 2022-12-09
> **Summary of Paper Revision**
>
> Dear Reviewer Rh4z,
>
> Thank you again for giving us many constructive comments!
>
> Since the discussion period is closing soon, we truly appreciate if you could check our responses and kindly let us know if you have any further feedback.
>
> We have revised our paper according to comments from all reviewers, and we would like to briefly summarize our revisions as follows.
>
> 1. [Abstract] We have rewritten part of the Abstract to clarify our contributions. In particular, we emphasize that “FairAC can be applied to various types of homogeneous graphs and generate fair embeddings for them”.
>
> 2. [Related Work] We further clarify the major differences between our FairAC and existing graph fairness methods (e.g., FairGNN) and graph attribute completion methods. Specifically, we show the differences from the perspectives of problem definition, methodology, and technical approaches.
>
> 3. [Section 3.1, Problem Definition] We refine the problem definition to make it clear. In the original version, we presented a simple case of binary node classification. In fact, our FairAC method could be easily extended to other graph learning tasks, such as multi-class node classification and regression.
>
> 4. [Tables] We have run additional experiments to show the mean and standard deviation values. We also updated the results in Table 2 based on the latest experiments.
>
> 5. [Sections 4.2 and 4.3] Instead of discussing our performance in a general way, we evaluate our method and show the improvement quantitatively. Our FairAC method can significantly improve the performance in terms of the fairness metric \deltaSP+\deltaEO, i.e., 65%, 87%, and 67% improvement over FairGNN on the NBA, pokec_z, pokec_n datasets, respectively.
>
> 6. [Appendix A.1] We added more information on implementation details. We set a threshold for accuracy to evaluate fairness, because there is a trade-off between accuracy and fairness.
>
> Sincerely,
>
> Authors of Paper 946

---

### Official Review · Reviewer_ifZW · 2022-10-24

**Confidence:** 4
**Correctness:** 4
**Technical Novelty And Significance:** 2
**Empirical Novelty And Significance:** 2
**Recommendation:** 6

**Clarity, Quality, Novelty And Reproducibility:**

Clarity: The writing is ok, but needs more comparison with FairGNN.
Quality: The empirical results are not that powerful.

**Strength And Weaknesses:**

Strength:
- The topic of fair attribute completion on graphs with missing attributes is relevant and natural.
- The pre-processing idea has the potential to be applied to multiple downstream tasks to improve their fairness performance.

Weaknesses:
- The novelty of the problem has not been clarified clearly. The authors mentioned that FairGNN [Dai & Wang (2021)] also involves the missing attribute problem, and claimed that the paper only assumes that a part of the sensitive attributes is missing. However, I don't see many differences between partial missing and full missing for nodes. What makes the full missing setting more challenging? What is the technical difficulty of proposing methods under this setting compared to the partial missing setting? These problems should be discussed more.
- The empirical results seem not that powerful to me. The performance of FairAC is quite close to FairGNN, sometimes better and sometimes worse with slight differences.

**Summary Of The Paper:**

The paper considers the problem of fair attribute completion on graphs with missing attributes. The authors proposed the FairAC framework, which jointly completes the missing features and mitigates unfairness. FairAC leverages the attention mechanism to complete missing attributes and adopts a sensitive classifier to mitigate implicit feature unfairness as well as topological unfairness on graphs. Experimental results on three real-world datasets demonstrate the superiority of the proposed FairAC framework over baselines in terms of both node classification performance and fairness performance.

**Summary Of The Review:**

The paper considers the problem of fair attribute completion on graphs with missing attributes and proposes the FairAC framework. The novelty of the problem has not been clarified clearly, and the empirical results seem not that powerful compared to FairGNN.

Overall, I do not recommend acceptance.

%%%%%%%%%%%%%%%%%%%%%%%%%%%%%%%%%%
Thanks for the response. It addresses my questions and I raise the score.

---

> ### Author Response · Authors · 2022-11-17
> **Thank you for providing valuable feedback. We would like to address your concerns and questions.**
>
>
> We thank the reviewer for providing valuable feedback. We have revised our paper based on all your comments.
>
> ### Q1
>
> >The novelty of the problem has not been clarified clearly. The authors mentioned that FairGNN [Dai & Wang (2021)] also involves the missing attribute problem, and claimed that the paper only assumes that a part of the sensitive attributes is missing. However, I don't see many differences between partial missing and full missing for nodes. What makes the full missing setting more challenging? What is the technical difficulty of proposing methods under this setting compared to the partial missing setting?
>
>
>
> We have explained the significant differences between FairGNN and our work in our paper (Section 2.1, Page 3).
>
>
>
> 1.  Problem Settings. FairGNN assumes that only a limited number of nodes are provided with sensitive attributes. Different from FairGNN, our setting is more realistic and even more difficult. Besides sensitive attributes, the non-sensitive attributes of some nodes are also missing in our setting.
>
>
>
>
> 2.  Methodology. FairGNN focuses on learning fair GNNs for node classification, and only a limited number of nodes are provided with sensitive attributes. Different from FairGNN, our FairAC focuses on learning fair node embeddings that can be applied to many downstream tasks where the attributes (including sensitive ones) of some nodes are completely missing.
>
>
>
>
> Technically, existing work cannot effectively address our problem, because they need to recover the missing information first from the remaining features of the node. And the main difference between partial missing and full missing is that the partial missing nodes can be inferred by their own features directly, while full missing nodes can only rely on their neighborhoods which is more difficult to find the relation. For example, FairGNN adopts a sensitive classifier to predict the missing sensitive labels. After that, it employs a classic adversarial model to mitigate unfairness. Specifically, a sensitive discriminator aims to predict the known or estimated sensitive attributes, while a GNN model tries to fool the sensitive discriminator and meanwhile predicts node labels. However, FairGNN fails to predict sensitive information if the features of a node are completely missing in the first place, and thus it will fail to achieve their final goal. Our FairAC can get rid of the problem, because we recover the node embeddings from their neighbors.
>
>
>
> In addition, it is much more difficult to infer missing attributes from neighbors, because it is harder to learn causal relation [1] between one node and the neighbors than those between attributes in one node.
>
>
>
> Overall, although both the FairGNN and our work focus on fair learning on graphs, there are significant differences in terms of problem settings, technical challenges, and methodology.
>
>
>
> [1] Pearl, Judea. "Causal inference in statistics: An overview." Statistics surveys 3 (2009): 96-146.
>
>
>
>
> ### Q2
>
> >The empirical results seem not that powerful to me. The performance of FairAC is quite close to FairGNN, sometimes better and sometimes worse with slight differences.
>
>
>
> To evaluate fairness of compared methods, we follow the widely used evaluation protocol in fair graph learning and set a threshold for accuracy, because there is a trade-off between accuracy and fairness. Since we mainly focus on the fairness metric, we set the accuracy threshold that all methods can satisfy, e.g., around 66% on the pokec_n dataset . In Table 1 and Table 2, our method obtains the best fairness performance and can maintain a comparable performance on accuracy and AUC.
>
>
>
> Compared with the SOTA method FairGNN, our FairAC method can significantly improve the performance in terms of the fairness metric \deltaSP+\deltaEO, i.e., 65%, 87%, and 67% improvement over FairGNN on the NBA, pokec_z, pokec_n datasets, respectively.
>
> [1] Enyan Dai and Suhang Wang. Say no to the discrimination: Learning fair graph neural networks with limited sensitive attribute information. WSDM, 2021.

---

> ### Author Response · Authors · 2022-12-09
> **Thanks for your further comment!**
>
> Dear Reviewer ifZW,
>
> Thank you so much for checking our responses, and thanks again for your valuable comments!
>
> Sincerely,
>
> Authors of Paper 946

---

### Official Review · Reviewer_sBeK · 2022-10-24

**Confidence:** 4
**Correctness:** 3
**Technical Novelty And Significance:** 2
**Empirical Novelty And Significance:** 3
**Recommendation:** 6

**Clarity, Quality, Novelty And Reproducibility:**

The paper is relatively clear, although there are some areas that need polishing and some concepts (most notably, the risk of inference of sensitive information) require further evaluation. The quality of the ideas are interesting and the model is relatively parsimonious. The novelty is more with respect to the application of existing tools to solve a problem in attribute prediction.

**Details Of Ethics Concerns:**

I did not identify a direct violation.

**Strength And Weaknesses:**

With respect to the strengths, the paper is well organized, the problem is relevant to the ML community, particularly the graph-learning and GNN and network science communities. Clear goals despite some imprecisions in the definitions. Relative parsimonious model that deals with specific sub-goals to achieve the final aim of fairness in completion.
There are some areas where the paper could be improved. For instance, there are several imprecisions such as in the abstract where it is stated "FairAC can be applied to any graph and generate fair embeddings”. The evidence provided is solid but stating that it will generate fair embedding to any graph is too optimistic. The definition of matrix $\mathcal X$ is that it has a dimension of attributes $D$ yet each of the labels (and their predictions) are defined as a unidimensional $y \in{0,1}$. The problem definition is a little hand-wavy. What is a fair feature embedding? The concept of “fair attribute completion model”  relies on fair feature embedding and, although it is discussed in other paragraph that it is measured via Statistical Parity and Equal Opportunity, the concept is not formally defined. The feature unfairness description regarding how “some non-sensitive attributes could infer sensitive information” seems not to be measured by Statistical Parity (SP) and Equal Opportunity (EO). The former implies also a privacy concerned while SP and EO deal with fairness of representation. Finally, the statement "removing sensitive information implicitly contained in non-sensitive attributes" has also the implication that sensitive information removed could affect performance when the amount of datapoint from subjects belonging to a minority group is not well represented. How is that scenario handled?

**Summary Of The Paper:**

This paper presents a method (FairAC) to jointly addresses the graph attribute completion and graph unfairness problems due to both feature unfairness and topological unfairness due to missing attributes.
The model uses a set of $V_{keep}$ and $V_{drop}$ nodes as in Jin et al. (2001) and uses attribute completion with attention mechanism for nodes without features (this may introduce topological unfairness which   is handled by a sensitive classifier) and uses feature unfairness mitigation for nodes with attributes by removing implicit sensitive information that is present in non-sensitive attributes via an autoencoder. Fairness is measured through Statistical Parity and Equal Opportunity.
The experiments compare FairAC with several baselines on three datasets. It tests node classification accuracy, fairness w.r.t. sensitive attributes, and effects of sensitive attributes, missing rates, and ablations (attribute missing rate, adversarial training loss relevance, and the tradeoff hyper parameter $\beta$).

**Summary Of The Review:**

The paper has its pros, such as the relevance, the clear stated goals and evidence that seem to support the claim and the parsimonious solution to the complex problem. However, there are several imprecisions and some statements that seem to be not directly supported by the evaluation, such as the privacy statement on inference of sensitive information.

%%%Post Rebuttal Comment%%%
Thank you for the answer to my questions. This clarifies my doubts and raise my score.

---

> ### Author Response · Authors · 2022-11-17
> **Thank you for providing valuable feedback to improve our paper. We would like to address your concerns and questions**
>
>
> We thank the reviewer for providing valuable feedback to improve our paper. We have revised our paper based on all your comments.
>
> ### Q1
>
> >The evidence provided is solid but stating that it will generate fair embedding to any graph is too optimistic.
>
> We agree with the reviewer that the claim on generating fair embeddings to any graph is quite optimistic. In our paper, we have revised this sentence as: “FairAC can be applied to various types of homogeneous graphs and generate fair embeddings for them”. In fact, experiments on multiple homogeneous graphs show that our FairAC method can generate more effective fair embeddings than baselines.
>
> ### Q2
>
> > The definition of matrix X is that it has a dimension of attributes D yet each of the labels (and their predictions) are defined as a unidimensional y∈0,1
>
> Thank you for your valuable comment. Here we would like to provide some clarifications on our problem definition. When we describe the problem definition, we present a simple case of binary node classification. In fact, our FairAC method could be easily extended to other graph learning tasks, such as multi-class node classification and regression. We have revised the problem definition in our paper.
>
> ### Q3
>
> >What is a fair feature embedding? The feature unfairness description regarding how “some non-sensitive attributes could infer sensitive information” seems not to be measured by Statistical Parity (SP) and Equal Opportunity (EO). The former implies also a privacy concerned while SP and EO deal with fairness of representation.
>
>
>
> Fair embedding in this case refers to embeddings that contain no sensitive information. We agree with the reviewer that SP and EO cannot explicitly measure how “some non-sensitive attributes could infer sensitive information”, but we think they can still provide meaningful evaluations in an implicit way. In fact, we can evaluate fairness only in downstream tasks. Sensitive information can be described as a privacy property, while SP and EO are used as fairness metrics. We have an adversarial approach to help the encoder generate embeddings that cannot infer their sensitive information. In this case, there is no difference between privacy concerns and fairness concerns, because they all aim to mitigate affection of sensitive information.
>
>
>
> ### Q4
>
> >the statement "removing sensitive information implicitly contained in non-sensitive attributes" has also the implication that sensitive information removed could affect performance when the amount of datapoint from subjects belonging to a minority group is not well represented. How is that scenario handled?
>
>
>
> Yes. Removing some sensitive information will cause performance drop. However, the majority group will benefit from their sensitive property while minority group may have a little or no benefit from their sensitive property. Thus, if we disentangle their sensitive information and non-sensitive ones, the model is encouraged to make fair predictions. Strategies like graph augmentation might help address this issue, which will be explored in our future work.

---

> ### Author Response · Authors · 2022-12-09
> **Thanks for your post-rebuttal comment!**
>
> Dear Reviewer sBeK,
>
> Thank you very much for checking our responses, and thanks again for giving us many constructive comments, which are truly helpful in improving our paper!
>
> Sincerely,
>
> Authors of Paper 946

---

### Official Review · Reviewer_ntMx · 2022-11-03

**Confidence:** 2
**Correctness:** 2
**Technical Novelty And Significance:** 2
**Empirical Novelty And Significance:** 2
**Recommendation:** 5

**Clarity, Quality, Novelty And Reproducibility:**

This paper focuses on tackling unfairness in graph learning models with fair novelty and quality.


**Strength And Weaknesses:**

The author claims that FiairAC is the first method to jointly solve the problem of graph attribute completion and graph unfairness.

**Summary Of The Paper:**

In practice, due to the lack of data or privacy issues, the attributes of some nodes may not be accessible, which makes fair graph learning more challenging. This paper proposes a fair attribute completion method FairAC to supplement the missing information, learn the fair node embedding of the missing attribute graph, and adopt a attention mechanism to deal with the attribute missing problem to improve their performance.
The author claims that FiairAC is the first method to jointly solve the problem of graph attribute completion and graph unfairness. The experimental results show that this method has lower sacrifice and better fairness in fair learning.

**Summary Of The Review:**

My research field is not related to this manuscript and I am not very familiar with the research content.

---

> ### Author Response · Authors · 2022-11-17
> **Thank you very much for your review. We have revised our paper based on the comments from all reviewers.**
>
> Thank you very much for your review. We have revised our paper based on the comments from all reviewers.

---

> ### Author Response · Authors · 2022-12-09
> **Gentle Reminder of Further Feedback**
>
> Dear Reviewer ntMx,
>
> Thank you very much for reviewing our paper and giving us your initial comments. We have provided detailed responses to every comment/question from all reviewers and have also revised our paper accordingly.
>
> Since the discussion period is closing soon, we truly appreciate if you could check our responses as well as the revised paper, and kindly let us know if you have any further feedback.
>
> Thank you very much!
>
> Sincerely,
>
> Authors of Paper 946

---

### Author Response · Authors · 2022-11-28
**Thanks for all reviewers**

We are grateful to all the reviewers for their insightful comments and constructive suggestions. We have revised our paper based on their comments with major changes highlighted in blue color.

In addition, to reproduce our results, we shared the source code of our FairAC with usage description anonymously in this link: https://drive.google.com/file/d/1TO4W9dFnljQZiTTJl_kVkmTdmCwsj4lE/view?usp=sharing

Thank you for your reviews again!

---

### Decision · Program_Chairs · 2023-01-20

**Decision:**

Accept: poster

**Justification For Why Not Higher Score:**

This work simply does not rise to the level of oral/spotlight: it is borderline.

**Justification For Why Not Lower Score:**

This approach to fairness seems natural and important: missing/inaccessible attributes is a very common issue.

**Metareview: Summary, Strengths And Weaknesses:**

The paucity of reliable data and privacy issues may render some attributes of nodes in graph data inaccessible, which is a challenge for fair graph learning. This paper develops FairAC, a fair attribute-completion method to learn a fair node embedding of a given graph. FairAC adopts an attention mechanism to deal with missing attributes; it is developed to mitigate two types of unfairness---feature unfairness from attributes and topological unfairness due to attribute completion. Experimental results show that this method may have better fairness in learning.

To evaluate the fairness of the methods compared, the authors adopt a widely-used evaluation protocol in fair graph learning and set a threshold for accuracy, since there is a trade-off between accuracy and fairness. Since the primary focus is fairness, the authors set the accuracy threshold that all methods can satisfy, e.g., around 66% on the pokec_n dataset . Tables 1 and 2 suggest that this work obtains the best fairness performance and can maintain a comparable performance on accuracy and AUC. Specifically, in comparison with the SOTA approach of FairGNN, FairAC is shown to improve performance in terms of the fairness metric \deltaSP+\deltaEO: 65%, 87%, and 67% improvement over FairGNN on the NBA, pokec_z, and pokec_n datasets respectively.

The authors are also asked to improve the presentation as suggested by the referees.


**Note From Pc:**

if the above contains the word "oral" or "spotlight" please see: "oral" presentation means -> notable-top-5% and "spotlight" means -> notable-top-25%. As stated in our emails, we are disassociating presentation type from AC recommendations

**Summary Of Ac-Reviewer Meeting:**

I read the paper myself and have come to a decision largely similar to the referees'.